# Wheat Straw Return Influences Soybean Root-Associated Bacterial and Fungal Microbiota in a Wheat–Soybean Rotation System

**DOI:** 10.3390/microorganisms10030667

**Published:** 2022-03-21

**Authors:** Hongjun Yang, Yao Zhao, Jiaxin Ma, Zhenyang Rong, Jiajia Chen, Yuanchao Wang, Xiaobo Zheng, Wenwu Ye

**Affiliations:** 1Key Laboratory of Plant Immunity, Department of Plant Pathology, Nanjing Agricultural University, Nanjing 210095, China; hjyang@jsafc.edu.cn (H.Y.); yaozhao@njau.edu.cn (Y.Z.); 2016802186@njau.edu.cn (J.M.); zhenyangrong@163.com (Z.R.); jiajiachen@jsafc.edu.cn (J.C.); wangyc@njau.edu.cn (Y.W.); xbzheng@njau.edu.cn (X.Z.); 2Key Laboratory of Soybean Disease and Pest Control (Ministry of Agriculture and Rural Affairs), Nanjing Agricultural University, Nanjing 210095, China; 3College of Agronomy and Horticulture, Jiangsu Vocational College of Agriculture and Forestry, Zhenjiang 212400, China; 4College of Landscape Architecture, Jiangsu Vocational College of Agriculture and Forestry, Zhenjiang 212400, China

**Keywords:** microbiome, bulk soil, rhizosphere, rhizoplane, endosphere, soybean root, wheat straw return, plant pathogen

## Abstract

Roots hold complex microbial communities at the soil–root interface, which can affect plant nutrition, growth, and health. Although the composition of plant microbiomes has been extensively described for various plant species and environments, little is known about the effect of wheat straw return (WSR) on the soybean root microbiota. We used Illumina-based 16S rRNA and ITS amplicon sequencing to track changes in bacterial and fungal microbiota in bulk soil and soybean rhizosphere, rhizoplane, s1and endosphere during the third and fourth years after implementing WSR in a wheat–soybean rotation system. The results revealed that WSR had a greater impact on fungal communities than bacterial communities, particularly in bulk soil, rhizosphere, and rhizoplane. WSR enriched the relative abundance of cellulose-degrading fungi (e.g., *Acremonium*, *Trichoderma*, and *Myrmecridium*, among which *Trichoderma* also had antimicrobial activity), saprotroph (e.g., *Exophiala*), and nitrogen cycling bacteria (e.g., *Chryseolinea*). Furthermore, WSR depleted the relative abundance of pathogenic fungi (e.g., *Fusarium* and *Alternaria*). These data revealed for the first time that WSR had diverse effects on soybean root-associated microbial community composition, not only in soil but also in the rhizosphere, rhizoplane, and endosphere.

## 1. Introduction

The roots of soil-grown plants are the main site of interactions with soil microbes and are in direct proximity to the largest known reservoir of microbial diversity [1,2]. Plants rely on interactions between roots and microbes for various beneficial effects, such as improved stress resistance, improved nutrient availability, promoted plant growth, and suppressed pathogen infection [2,3,4,5]. Roots have distinct compartments (rhizosphere (soil close to the root surface (RS)), rhizoplane (root surface (RP)), and endosphere (root interior (ES))) with varying microbial diversity at the soil–root interface [2,4,6,7].

Wheat straw return (WSR), i.e., wheat stalk, was crushed into pieces and added directly to the soil surface. WSR provides a valuable source of carbon for soil microbes, and a series of studies have investigated the effects of WSR on soil microbial communities using high-throughput sequencing. For example, wheat–rice or wheat–maize straw return boosted the variety of soil bacterial and fungal communities substantially [8,9,10,11]. In a wheat–soybean rotation system, WSR also altered nitrogen cycling and pathogen-associated soil microbiota [12]. Rice straw return has also been shown to have an impact on rhizosphere microbial communities of subsequently planted maize [13]. However, little is known about how WSR affects crop-root-associated microbiomes, which are more intimately linked to plant health than field soil.

The current study consisted of a two-season field experiment conducted in 2017 and 2018 in a wheat–soybean cropping system in three sites of the Huang-Huai region of China. We compared differences in bacterial and fungal communities among the RS, RP, and ES of soybean roots, and bulk soil (BS) from soybean fields, and evaluated the influence of WSR upon soybean root-associated microbiota. We hypothesized that WSR could not only affect the microbial community in BS but also in RS, RP, and ES. Further, some nitrogen cycling bacteria and pathogenic fungi would also be affected by WSR. The results are expected to facilitate the implementation of effective agricultural microbial community management strategies and the development of sustainable agriculture.

## 2. Materials and Methods

### 2.1. Field Experiment Design

A two-season field trial was conducted in 2017 and 2018 in a wheat–soybean cropping system in three sites of the Huang-Huai region of China, including Jining, Shandong Province (35°27′ N, 116°35′ E, sandy loam soil); Xuzhou, Jiangsu Province (34°17′ N, 117°17′ E, yellow loam sand soil); and Suzhou, Anhui Province (33°38′ N, 117°05′ E, mortar black soil). The system has been previously described by Yang et al. (2019) [12]. At each site, we sampled from two neighboring fields with two straw regimes: (1) no wheat straw return (NSR), i.e., artificial harvest of wheat, with wheat stubble less than 5 cm in height and all wheat straw removed from the field; and (2) wheat straw return (WSR), i.e., all wheat stalk was crushed into pieces and added to the soil surface without tillage after wheat harvest every season.

The wheat–soybean cropping system dominates all three sites, with winter wheat seeded in early October and harvested in early June, and summer soybean seeded in mid-June and harvested in late September. The wheat cultivar is Jimai 22 (high-yield cultivar), and the soybean cultivar is Zhonghuang13 (high-protein cultivar); both are popular in the Huang-Huai region of China. Wheat straw was applied at approximately 5000 kg·hm^−2^. The detailed experimental design, agronomic management, and fertilization regimes were consistent with those described by Yang et al. (2019) [12]. In brief, the plots received basal fertilization of 50 kg N-P_2_O_5_-K_2_O ha^−1^ and 15 kg urea ha^−1^ at wheat seeding, and 10 kg N-P_2_O_5_-K_2_O ha^−1^ was applied at soybean seeding, and the three sites were consistent. The years 2017 and 2018 were the third and fourth years after implementing wheat straw return.

The indoor seed test and yield determination were carried out after soybean ripening. Ten consecutive plants with normal growth in the middle row were taken from each plot for seed test, and the plant height, bottom pod height, number of main stem nodes, effective branch number, effective pod number, grains number per plant, grain weight per plant, and 100-grain weight were investigated. Six rows were harvested manually in each plot, each row 3 m, with a total area of 7.2 m^2^, and the yield was calculated by threshing.

### 2.2. Sample Collection

Samples were collected in mid-August (60 days after soybean planting) of each year and included four compartments associated with soybean roots (bulk soil, rhizosphere soil, rhizoplane, and endosphere). In total, 240 samples were collected (3 sites × 4 compartments × 2 treatments × 5 subplots per treatment × 2 years). All samples were stored at −20 °C until DNA extraction.

For collection of bulk soil samples, ten subplots (five with wheat straw return and five with no wheat straw return; 480 m^2^ in total) were chosen, and nine cores (20 cm in depth) were collected from each subplot with shovels (3.8 cm diameter) using an S-formation sampling method (Appendix A). The soils from nine cores of each subplot were mixed to form one composite sample. The soil samples were placed into separate sterile plastic bags and transported to the laboratory on ice. Bulk soil samples were consistent with the description of Yang et al. (2019) [12], i.e., each soil sample was sieved through a 2 mm mesh filter to remove roots and plant detritus.

For collection of samples representing the rhizosphere, rhizoplane, and endosphere compartments of roots, nine soybean plants from each subplot (with an S-formation sampling method) were pooled to form one composite sample (Appendix A). Sampling of compartment separations was carried out following a previously reported protocol with some modifications [4], as described below.

The rhizosphere soil was strictly defined as soil particles adhering to the roots. Excess soil was manually shaken from the roots. The roots of each plant were cut into segments of 5–7 cm, and 5 g of each composite sample was analyzed. We separated the soil from the roots by placing the roots with soil still attached in a sterile beaker with 50 mL of sterile phosphate-buffered saline (PBS). The roots were stirred vigorously with sterile forceps to clean the soil from their surfaces. The soil that was cleaned from the roots was poured into a 50 mL Falcon tube and stored as the rhizosphere compartment.

After removing the rhizosphere soil, the roots were placed into a new 50 mL Falcon tube with 15 mL PBS, and tightly adhering microbes at the root surface were removed using a sonication protocol. The roots in the Falcon tube were sonicated for 30 s at 50–60 Hz (output frequency, 40 kHz; power, 500 W). The roots were then removed, and the liquid PBS fraction prepared by centrifugation was kept as the rhizoplane compartment.

The root treated by sonication was then placed into a new 50 mL Falcon tube for two more sonication procedures using clean PBS solution (as described above). To ensure that all epiphytic microbes were cleared from the root surface, root compartments were then supplemented with 50 mL distilled water and two drops of Tween-20 at 25 °C, then shaken at 220 rpm for 20 min. Finally, surface sterilization was performed following previously described protocols [14] with some modifications. Surfaces were sterilized with sterile water (20 s), 70% (*v*/*v*) ethanol (30 s), and 2.5% (*v*/*v*) sodium hypochlorite (2 min), and then rinsed three times with sterile water. The sterilized roots were kept as the endosphere compartment.

### 2.3. DNA Extraction and Amplicon Sequencing

The rhizosphere soil was concentrated by centrifuging for 30 s at 10,000× *g*. The supernatant was discarded, leaving only the soil fraction behind. The rhizoplane compartment was concentrated in the same manner. Approximately 250 mg of bulk soil, rhizosphere soil, and rhizoplane samples was used for each individual DNA extraction. DNA was extracted using the MoBio PowerSoil DNA Isolation Kit according to the manufacturer’s instructions (MoBio Laboratories Inc., Carlsbad, CA, USA). For the root tissues, DNA was extracted using the DNAsecure Plant Kit (Tiangen, Beijing, China) according to the manufacturer’s instructions. DNA concentration and purity were quantified using a NanoDrop 1000 Spectrophotometer (Thermo Fisher Scientific, Waltham, MA, USA), and DNA quality was checked by 1% agarose gel electrophoresis.

PCR amplification and library preparation were performed following previously published (Yang et al., 2019) [12]. The 16S rRNA and ITS1 genes were amplified using the primers 515F:806R [4,7,13,15] and ITS1-F:ITS2 [7,12,13,14,16], respectively. For 16S rRNA gene libraries, PCR reactions were performed in triplicate 20 μL mixture containing 4 μL of 5 × FastPfu Buffer, 2 μL of 2.5 mM dNTPs, 0.8 μL of each primer (5 μM), 0.4 μL of FastPfu Polymerase, 0.2 μL of BSA, and 10 ng of template DNA. For fungal libraries, PCR reactions were performed in a triplicate 20 μL mixture containing 2 μL of 10 × buffer, 2 μL of 2.5 mM dNTPs, 0.8 μL of each primer (5 μM), 0.2 μL of rTaq Polymerase, 0.2 μL of BSA, and 10 ng of template DNA. The PCR reactions were carried out with the following program: an initial phase consisting of 95 °C for 3 min, followed by 27 cycles for 16S and 35 cycles for ITS of 95 °C for 30 s, 55 °C for 30 s, and 72 °C for 45 s, followed by a final extension at 72 °C for 10 min. The resulting PCR products were extracted from 2% agarose gel and further purified using the AxyPrep DNA Gel Extraction Kit (Axygen Biosciences, Union City, CA, USA) and quantified using QuantiFluor™-ST (Promega, WI, USA) according to the manufacturer’s protocol.

### 2.4. Analysis of Sequencing Data

Purified amplicons were pooled in equimolar and paired-end sequenced (2 × 300) on an Illumina MiSeq platform (Illumina, San Diego, CA, USA) according to the standard protocols by Majorbio Bio-Pharm Technology Co. Ltd. (Shanghai, China). Data processing referred to Yang et al. (2019) [12]. Operational taxonomic units (OTUs) were clustered with 97% similarity cutoff using UPARSE (v. 7.0.1090) (http://www.drive5.com/uparse/) (accessed on 18 May 2020) [17] after chimeras were filtered out using UCHIME (v. 4.2.40) [18]. OTUs were classified using a confidence threshold of 0.7 with the Silva database (v. 128) (https://www.arb-silva.de/) (accessed on 18 May 2020) [19] and UNITE database (v. 8.0) (https://unite.ut.ee/) (accessed on 18 May 2020) [20] for bacterial and fungal communities, respectively. OTUs not assigned to target species were removed from the data set.

OTUs classified as “Cyanobacteria” and “Mitochondria” were discarded from the OTU table for 16S rRNA genes, and those classified as “norank” and “unclassified_k__Fungi” were filtered out from the OTU table for ITSs before analysis. We obtained approximately 49,486 and 43,571 high-quality sequence reads per sample for the 16S rRNA gene and the ITS1 region, respectively. According to the minimum total number of reads among all samples, 30,079 and 30,083 sequence reads were rarefied per sample for the 16S rRNA gene and the ITS1 region, respectively. Fungal and bacterial α-diversities were estimated by calculating the Shannon diversity indices in Usearch (v. 7.0 http://drive5.com/uparse/) (accessed on 18 May 2020). Principal coordinates analyses (PCoAs) of the Bray–Curtis distances were performed using the “pcoa” function in the R package (https://www.r-project.org) (accessed on 18 May 2020). Analysis of similarities (ANOSIM) based on Bray–Curtis distances was performed to evaluate the significant differences between NSR and WSR using the vegan package.

### 2.5. Statistical Analyses

The relative abundances of bacteria and fungi, α-diversity indices, cellulose-degrading fungi, and plant pathogen-associated fungi between NSR and WSR across different compartments and sites were compared using Wilcoxon rank-sum tests (*p* < 0.05). The relative abundances of the most abundant bacterial and fungal higher taxa in each compartment were compared with one-way analysis of variance (ANOVA), followed by Tukey’s HSD (*p* < 0.05). The indoor seed test and yield of soybean between NSR and WSR were compared using Student’s *t*-test (*p* < 0.05). To identify phylum and genera that were significantly impacted by WSR, the datasets were subsampled according to sampling site, year, and compartment, and significant differences between NSR and WSR were tested with the Wilcoxon rank-sum test (*p* < 0.05). All statistical analyses were performed using SPSS v. 20.0 (SPSS Inc., Chicago, IL, USA).

## 3. Results

### 3.1. Effects of Wheat Straw Return on Plant Height and Yield of Soybean in Three Sites

The indoor seed test and yield of soybean for the treatments of NSR and WSR are summarized in Appendix A. The bottom pod height, effective branch number, and effective pod number had no significant differences between NSR and WSR across three sites in two years. However, the 100-grain weight of WSR was higher than that of NSR in the three sites within two years, and there was a significant difference (Student’s *t*-test, *p* < 0.05) in Jining in 2017. Furthermore, the plant height of WSR also increased in Jining in two years, and there was a significant difference (Student’s *t*-test, *p* < 0.05) in 2018 (Appendix A).

### 3.2. Diversity of Root-Associated Microbial Communities Was Moderately Affected by Wheat Straw Return

To assess how bacterial and fungal communities were affected by the wheat straw treatment, two different treatments (NSR and WSR) and three different sites (JN, SZ, and XZ) were considered. Across the three root-associated compartments (RS, RP, and ES) and BS, we characterized the bacterial and fungal communities in 240 samples by sequencing the amplicon of the V4 region of the 16S rRNA gene and the internal transcribed region 1 (ITS1), respectively. In total, 43 phyla, 116 classes, 233 orders, 465 families, and 964 genera were annotated from the 10,905 bacterial OTUs, and 7 phyla, 29 classes, 95 orders, 207 families, and 437 genera were annotated from the 1725 fungal OTUs.

The results showed that bacterial and fungal communities under the two treatments (NSR and WSR) exhibited similar α diversity, as estimated by Shannon’s diversity index in the four compartments (BS, RS, RP, and ES; Wilcoxon rank-sum test, *p* > 0.05). Straw return has an inconsistent impact on root-associated microbial diversity across sites (Jining (JN), Suzhou (SZ), and Xuzhou (XZ)), years (2017 and 2018), and compartments (Figure 1A,B). For example, relatively lower bacterial diversity was observed in the WSR RS than in the NSR RS in JN and XZ, while the opposite effect was observed in SZ; however, the difference in α-diversity between the two treatments did not reach the level of statistical significance (Wilcoxon rank-sum test, *p* > 0.05) in 2017 or 2018 (Figure 1A). Relatively higher fungal diversity was observed in WSR RS than in NSR RS in JN and XZ (Wilcoxon rank-sum test, *p* < 0.05; 2017 in XZ), while the opposite effect was observed in SZ (Wilcoxon rank-sum test, *p* < 0.05; 2018) (Figure 1B).

### 3.3. Root-Associated Fungal Community Composition Was More Influenced by Wheat Straw Return than Bacterial Communities

Based on principal component analysis (PCoA) and analysis of similarities (ANOSIM), we found significant differences (*p* < 0.001) in both bacterial and fungal community composition, across the four compartments (BS, RS, RP, and ES), the three sites (JN, SZ, and XZ), and the two years (2017 and 2018) (Figure 2A,B). For bacterial community composition, the impact of compartment was much higher than that of site or year (compartment R = 0.614, *p* < 0.001, site R = 0.242, *p* < 0.001, year R = 0.072, *p* < 0.001; Figure 2A). For fungal community composition, the impact of compartment and site was equal and higher than that of year (compartment R = 0.362, *p* < 0.001, site R = 0.385, *p* < 0.001, year R = 0.130, *p* < 0.001; Figure 2B). The impact of WSR on both bacterial and fungal community composition was weaker than each of the abovementioned three factors. The impact of WSR on the fungal community composition (R = 0.031, *p* < 0.001) was significant and stronger than that on the bacterial community composition (R = 0.002, *p* > 0.05) (Figure 2A,B).

Among the four compartments, WSR had a minor impact on bacterial community composition, except for SZ BS in 2017 (R = 0.280, *p* < 0.05), XZ RS in 2018 (R = 0.208, *p* < 0.05), and SZ ES in 2017 (R = 0.456, *p* < 0.01) (Figure 3A and Appendix A). WSR had a strong impact on fungal community composition in BS and the soybean RS at all three sites, especially in SZ (BS-2017: R = 0.416, *p* < 0.01; BS-2018: R = 0.608, *p* < 0.05; RS-2017: R = 0.381, *p* < 0.01; RS-2018: R = 0.750, *p* < 0.05) and XZ (BS-2017: R = 0.492, *p* < 0.01; BS-2018: R = 0.296, *p* < 0.05; RS-2017: R = 0.916, *p* < 0.01; RS-2018: R = 0.428, *p* < 0.05) in both 2017 and 2018 (Figure 3B and Appendix A). Compared with BS and RS, composition of fungal communities living on and within soybean roots (RP and ES) were relatively less affected by WSR (Figure 3B and Appendix A).

### 3.4. Phylum-Level Stability of Root-Associated Bacterial and Fungal Communities under Wheat Straw Return

Since 63.7% and 78.6% of the OTUs in the 16S and ITS data set were identified as Proteobacteria phylum and Ascomycota phylum, respectively, we further split these groups into their respective classes. There were similar relative abundances of bacterial and fungal phyla (or classes) between NSR and WSR treatments across the bulk soil and root-associated compartments at all three tested sites but notable differences among the bulk soil and root compartments (Figure 4A,B).

Twelve dominant (≥1.0% relative abundance) bacterial phyla and Proteobacteria classes were detected (Figure 4A). The ES had a significantly greater proportion of Alphaproteobacteria than the RS, RP, or BS, whereas Acidobacteria, Chloroflexi, Gemmatimonadetes, Verrucomicrobia, and Firmicutes were mostly depleted in the ES compared with the BS, RS, or RP (ANOVA and Tukey’s HSD, *p* < 0.05) (Figure 4A). The decrease in the relative abundance of these phyla across the root compartments is consistent with the observation that bacterial diversity decreases gradually from the bulk soil to the endosphere (Figure 1A). Eleven dominant (≥1.0% relative abundance) fungal phyla and Ascomycota classes were found. The relative abundance of Zygomycota decreased from BS to ES; however, there was no clear trend for other fungal phyla (or classes) across the BS and root compartments (Figure 4B).

When considering the relative abundance of bacterial and fungal phyla (or classes) that responded to WSR among the different sites and years of each compartment, we identified a total of nine bacterial and seven fungal phyla (or classes), which showed significantly differential abundance between NSR and WSR in at least one site or year within a specific compartment (Appendix A). To better reveal the WSR-affected differential abundance of microbial taxa and the functions of associated microbial taxa, we then performed genus-level analyses.

### 3.5. Differential Abundance of Bacterial Taxa under Wheat Straw Return

When considering the relative abundance of bacterial and fungal genera that responded to WSR among the different sites and years of each compartment, we identified a total of 59 bacterial genera, which showed significantly differential abundance between NSR and WSR in at least one site or year within specific compartment (Wilcoxon rank-sum test, *p* < 0.05). Among the four compartments, the number of WSR-responsive genera increased from BS (46) to RS (60) and then decreased to RP (40) and ES (26) (Appendix A). When further considering the consistency of changing trend among the different sites and years of each compartment, we identified 21 bacterial genera, which showed not only significant response to WSR in at least one site or year but also similar changing trend in at least five of the six cases (three site × two years for one compartment) (Figure 5A).

Among the 21 identified bacterial genera, which showed a relatively consistent changing trend under WSR, 15 genera showed an increasing trend, and the number was more than those showing a reducing trend (6 genera) (Figure 5A). There were eleven genera (52%) having specific functions. Five genera were associated with nitrogen cycling, including *Chryseolinea* (BS and RS), *Pseudoduganella* (BS), *Ensifer* (RP), and *Nitrospira* (RP), which exhibited an increasing trend within varied compartments, and *Bradyrhizobium*, which exhibited a reducing trend in ES. *Massilia* associated with phosphate solubilization had a reducing trend in BS and RS. *Mycobacterium* associated with hydrocarbon degradation and *Acidibacter* containing acidophilic bacteria were both increased in BS. In addition, *Candidatus Solibacter* and *Bacillus*, associated with wheat and soybean diseases, were reduced in RS and RP, respectively, while *Lysobacter* associated with antimicrobial activity was increased in RP (Figure 5A).

### 3.6. Differential Abundance of Fungal Taxa under Wheat Straw Return

We identified a total of 45 fungal genera which showed significantly differential abundance between NSR and WSR in at least one site or year within a specific compartment (Wilcoxon rank-sum test, *p* < 0.05) (Appendix A). With respect to the fungal genera (16 in total), which showed a relatively consistent changing trend under WSR, there were also more genera showing an increasing trend than those showing a reducing trend (10 > 6) (Figure 5B). Among the 16 genera, 75% (12 genera) contained reported plant pathogens (termed plant pathogen-associated genera), and in ten genera, the contained plant pathogen species were reported to infect soybean and/or wheat (Figure 5B).

Among the 12 identified plant pathogen-associated fungal genera, five genera, i.e., *Aspergillus* (BS and RS), *Fusarium* (BS and RP), *Alternaria* (BS), *Corynespora* (BS), and *Penicillium* (BS), showed a consistent reducing trend under WSR across the compartments (Figure 5B and Figure 6A), and six genera, i.e., *Acremonium* (BS, RS, and RP), *Mycosphaerella* (BS and RP), *Pyrenochaetopsis* (BS and ES), *Pyrenophora* (BS), *Trichoderma* (RS and RP), and *Phaeosphaeria* (RS and ES), showed a consistently increasing trend under WSR within varied compartments (Figure 5B and Figure 6B). The *Trichoderma* genus, which showed an increasing trend, was also well known for its antimicrobial activity and cellulose-degrading activity (Figure 5B and Figure 6C). In addition, eight identified fungal genera are known to be associated with cellulose degradation, and six genera (e.g., *Myrmecridium* in RS and RP and *Podospora* in RS) showing a consistently increasing trend under WSR, while two genera (e.g., *Preussia* in RS; and *Fusarium* in BS and RP) were the opposite (Figure 5B and Figure 6C).

## 4. Discussion

Studies of plant microbiomes in different environments are an active area for research, but the work conducted to systematically characterize the root microbiome in soybean is still largely lacking [21,22]. Similarly, the effect of WSR on soybean root microbiota has not been previously investigated. This work provides the first detailed characterization of the short-term (third and fourth years) response of the soybean-associated microbial community to WSR in three locations in the Huang-Huai region of China. Our study demonstrated that the microbial community composition was significantly influenced by all these factors.

In order to extract high-quality and quantity DNA from bulk soil, rhizosphere, rhizoplane, and soybean root tissue samples, and reduce the deviation caused by low DNA recovery, we used two different kits to extract DNA. Although the use of different kits will affect the investigation of the microbiome to some extent, we cannot use one DNA extraction kit to extract high-quality and quantity DNA from all compartments. Previous research papers also used different kits to extract DNA from different samples. For example, Cregger et al. (2018) used MoBio PowerSoil DNA Isolation Kit (MoBio Laboratories, Inc., Carlsbad, CA, USA) to extract rhizosphere, phyllosphere, and bulk soil samples DNA and MoBio PowerPlant Pro DNA Isolation Kit (MoBio Laboratories, Inc., Carlsbad, CA, USA) to extract plant tissues samples DNA; Beckers et al. (2017) extracted rhizosphere soil DNA with Power Soil DNA Isolation Kit (MoBio, Carlsbad, CA, USA) and plant samples DNA with Invisorb Spin Plant Mini Kit (Stratec Biomedical AG, Birkenfeld, Germany) [23,24]. Nevertheless, our results show that the microbial community varies greatly between compartments. We evaluated the effects of WSR in different geographical locations, rhizocompartments, and years, as these factors are known to affect soil and rhizosphere microbial communities [13,15,25,26]. Our data suggest that microbial diversity and community composition in soybean rhizocompartments were moderately affected by WSR. Overall, community composition was most affected by root compartment and geographical location and least affected by straw return (Figure 2A,B). It was consistent with an earlier study in Italy and the Philippines, where straw treatment had a minor impact on soil and rhizosphere microbial community composition when compared to crop rotation, field location, niche (bulk soil or rhizosphere), and time. [13]. It is worth noting that in this study, we used UPARSE (v. 7.0.1090) (http://www.drive5.com/uparse/) (accessed on 18 May 2020) to cluster OTU based on 97% sequence similarity, which was called the “gold standard” of clustering methods of amplicon sequencing. This method not only simplified the workload but also improved the analysis efficiency. However, recently, a new method has been developed, that is, using DADA2 to cluster amplicon sequence variables (ASV) based on 100% sequence similarity [27]. However, ASV also has disadvantages. For example, some rich and very low species in the sample may be eliminated due to its more strict judgment method. Therefore, OTU clustering is still used in this study.

The rhizocompartments (bacterial R = 0.614, fungal R = 0.362) and geographical locations (bacterial R = 0.242, fungal R = 0.385) described the largest source of variation in the composition of microbial communities sampled. This conclusion is consistent with the fact that there are significant differences in the composition of microbial communities from different geographical and climatic regions. [7,13,25,28]. Moreover, root-associated compartments also significantly affected microbial community composition and diversity, as observed in previous studies [4,7]. The above-observed differences may be due to the variation in soil environmental factors, as well as the dynamic acquisition process of the root microbial community.

Compared with other factors, the short-term response to WSR was relatively mild. ANOSIM analysis showed that WSR had a stronger impact on fungal community composition than bacterial community composition, and the effects occurred mainly in the RP, RS, and BS (Figure 3A,B). Fungi play an important role in straw degradation, which can decompose more recalcitrant material, and decomposition products released by fungi may be an important nutrient source for bacteria [29,30,31,32]. This is consistent with a previous study showing that the effect of rice straw return on fungal communities in maize RS and BS was greater than the influence on bacterial communities [13]. Notably, the rhizosphere-priming effects can improve plant nutrition status by mineralizing nutrients released from refractory organic carbons [33]. These processes may lead to significant alterations in microbial community composition in the RS and RP after WSR.

Members of some of the WSR-enriched taxa (showing a consistently increasing trend) may belong to cellulose-degrading fungi. These genera have the ability to degrade plant residue [30,34,35,36,37,38,39,40,41]. Most of these fungal genera belong to the classes Sordariomycetes and Dothideomycetes, which is consistent with previous reports that genera belonging to the two classes were significantly enriched in Italian soil after rice straw return [13]. Furthermore, the relative abundance of *Pyrenochaetopsis* similarly increased following WSR. These fungi exhibit proteolytic activity and are also associated with elevated atmospheric carbon dioxide levels [8,42,43,44].

The relative abundance of some plant pathogen-associated fungal genera decreased after WSR, mainly in BS, RS, and RP, e.g., *Fusarium*, and *Alternaria*, while the relative abundance of *Mycosphaerella* increased (Figure 5B and Figure 6B,C). This is in line with our previous results, which revealed that the relative abundances of *Fusarium* and *Alternaria* were significantly lower in WSR soils, while that of *Mycosphaerella* was significantly higher [12]. These genera are known to contain many species of soil-borne soybean and/or wheat pathogens [45,46,47,48,49]. This effect may result from the presence of a large number of organic substrates in WSR treatment, which promote nutrient cycling and improve the ecological environment and nutrient status of soil and soybean root-associated microbial communities. It may also be related to the accumulation of antagonistic microbes after WSR, such as Trichoderma, a well-known biocontrol fungus. Such microbes could play an important role in the inhibition of soil-borne diseases caused by *Fusarium oxysporum* and *F. graminearum* [50,51].

For bacteria, more genera showed increased abundance under WSR, compared to those showing reduced abundance (Figure 5A). Among the enriched genera under WSR, we identified many nitrogen cycling-associated bacterial genera (e.g., *Chryseolinea*, *Pseudoduganella*, *Ensifer*, and *Nitrospira*) in BS, RS, or RP. This is in line with our previous findings, where the relative abundances of the nitrogen cycling-associated bacterial genera, such as *Nitrospira*, were significantly higher in soils treated with WSR within three years [12]. Thus, WSR exhibited an impact on the nitrogen cycling-associated microbiota in soil and distinct root-associated compartments.

## 5. Conclusions

Our study demonstrated that the effect of short-term WSR on the microbial community composition was different among different soybean root compartments, and the fungal community responded more strongly than the bacterial community. Many WSR-affected microbiota was associated with the functions such as cellulose degradation, nitrogen cycling, and plant infection, representing some major effects of short-term WSR on the soil ecology and plant health. Further investigations on the effects of long-term WSR on the microbial community, as well as the responses of the endophyte community of underground and aboveground organs of soybean to WSR would be interesting.

## Figures and Tables

**Figure 1 microorganisms-10-00667-f001:**
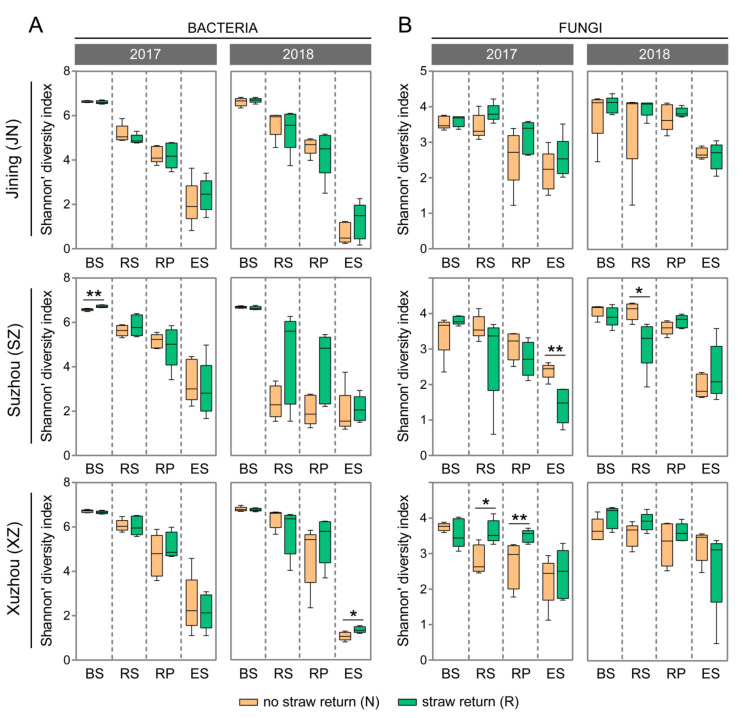
The influence of WSR on α-diversity (Shannon’s diversity index) for bacterial (**A**) and fungal (**B**) communities within each site, year, and compartment. Horizontal bars in boxes are medians; upper and lower box edges are the 75th and 25th quartiles, respectively; and whiskers are 1.5-fold the interquartile range. Based on Wilcoxon rank-sum test, significant differences between treatments within a compartment are indicated with “*” (*p* < 0.05) or “**” (*p* < 0.01).

**Figure 2 microorganisms-10-00667-f002:**
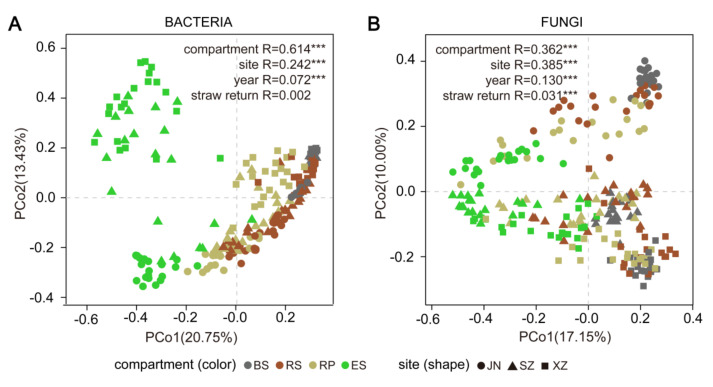
Principal coordinates analyses (PCoAs) showing the influence of compartment, site, year, and straw return on bacterial (**A**) and fungal (**B**) community composition based on Bray-Curtis dissimilarities. Analysis of similarities (ANOSIM) was applied to test for differences in community composition due to compartment, site, year, and straw return; R values are indicated with “***” for *p* < 0.001.

**Figure 3 microorganisms-10-00667-f003:**
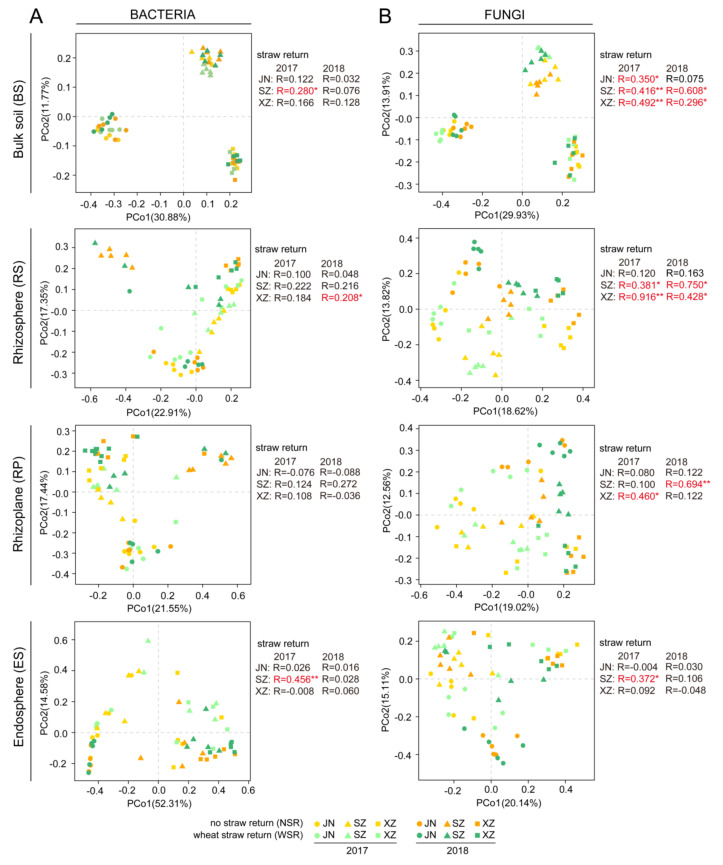
PCoAs showing the influence of WSR on bacterial (**A**) and fungal (**B**) community composition in bulk soil (BS), rhizosphere (RS), rhizoplane (RP), and endosphere (ES) based on Bray–Curtis dissimilarities. ANOSIM was applied to test for differences in community composition due to WSR across different compartments, sites, and years; R values are indicated with “*” for *p* < 0.05, “**” for *p* < 0.01, and marked the font in red.

**Figure 4 microorganisms-10-00667-f004:**
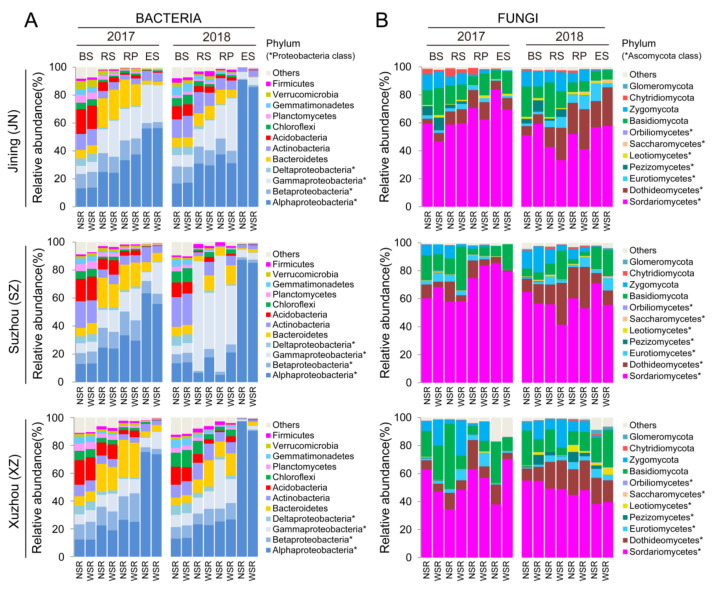
Effects of wheat straw return on the relative abundances of bacterial and fungal higher taxa. (**A**) Relative abundances of the most abundant (>1.0%) bacterial phyla and Proteobacteria classes (*) in each compartment, site, year, and straw treatment. (**B**) Relative abundances of the most abundant (>1.0%) fungal phyla and Ascomycetes classes (*) in each compartment, site, year, and straw treatment.

**Figure 5 microorganisms-10-00667-f005:**
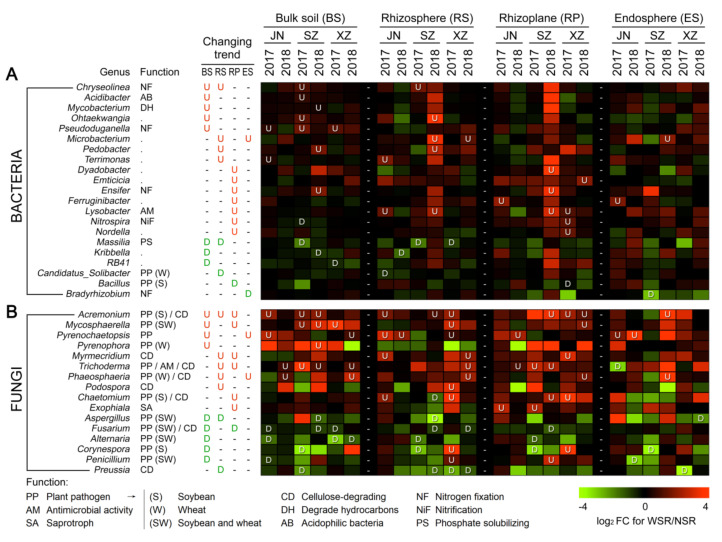
WSR-responsive genera of bacteria (**A**) and fungi (**B**) in each compartment, site, and year. The color of the heat map indicates the log_2_ fold change in relative abundance with respect to the control treatment, i.e., WSR/NSR: an increase tends toward red, while a decrease tends toward green. On the heap map, “U” and “D” mean that the relative abundance significantly increased and decreased, respectively (*p* < 0.05, Wilcoxon rank-sum tests). For the columns of “Changing trend”, “U” and “D” represent a consistently increasing and decreasing trend in the indicated compartment, respectively. A consistent changing trend was defined for a genus that showed not only a significant response to WSR in at least one site or year but also a similar changing trend in at least five of the six cases (three sites × two years for one compartment).

**Figure 6 microorganisms-10-00667-f006:**
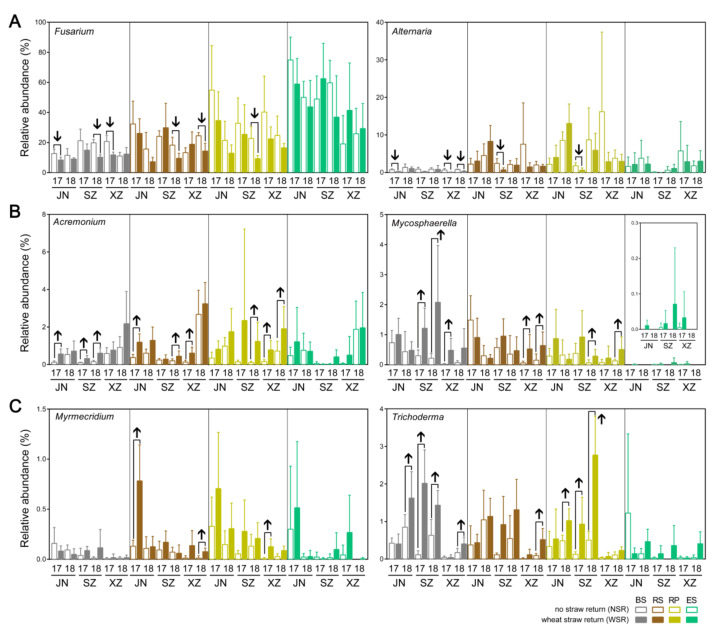
Relative abundance of the fungal genera which contain species reported as plant pathogen or associated with cellulose degradation. (**A**) *Fusarium* and *Alternaria*; (**B**) *Acremonium* and *Mycosphaerella*; (**C**) *Myrmecridium* and *Trichoderma*. Up and down arrows indicate significant increasing and reducing relative abundance due to WSR, respectively (*p* < 0.05, Wilcoxon rank-sum test).

## Data Availability

Raw sequence reads are available in the NCBI Sequence Read Archive (SRA) database under the accession number PRJNA587411.

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
