# Peer review of "Wheat Straw Return Influences Soybean Root-Associated Bacterial and Fungal Microbiota in a Wheat–Soybean Rotation System"

_microorganisms, 2022, doi:10.3390/microorganisms10030667_

Round 1

Reviewer 1 Report

Dear Authors,

The manuscript show influences cropping system soybean root-associated bacterial and fungal microbiota in a wheat–soybean rotation system. These results can be very interesting considering the implementation of effective agricultural microbial community management strategies and the development of sustainable agriculture.  The subject of this manuscript is consistent with the scope of the Journal. The conclusions corresponds with the work's content.

 Manuscript can be published in scientific Microorganisms after some changes (major revision):

  • It makes me wonder if using two different kits for isolation DNA could have affected the results ?
  • There is no information as to the use of repeats in DNA isolation. I wanted to ask in this regard if the DNA was isolated from only one small sample of about 250 mg. Is this not too small a sample?
  • I checked the information under the bio-project number PRJNA 587411. Unfortunately no such project exists. There are two possible guesses for this situation. The first, that the bioproject is not publicly visible and therefore I cannot check it. The second, that the bioproject does not exist. Please dispel my doubts.
  • Please, be sure that all the references cited in the manuscript are also included in the reference list and vice versa with matching spellings and dates.

Reviewer 2 Report

The paper entitled “Wheat Straw Return Influences Soybean Root-Associated Bacterial and Fungal Microbiota in a Wheat–Soybean Rotation System” reports an evaluation of the effect of WSR on microbial and mycobial communities in soil, rhizosphere, rhizoplane and endosphere.

The paper is well written, the experimental plan is justified  as well as the conclusions.

However, I think there are some changes that could improve the quality of the study. In fact there are some part of the study that need to be better discussed in the paper. These are:

  1. The fact that the endosphere samples were extracted with a different kit: this is known to bias the investigation of the microbiota. As you then compare these with the other samples, please discuss this in the paper.
  2. You used OTUs and not ASVs. OTUs are a bit old fashion now. (https://www.nature.com/articles/ismej2017119) I do suggest you comment on that in the discussion.

Finally, you compare 3 things: geography (3 locations), plant compartments and time. Sometimes it is difficult to read how these 3 things affect the community. I do suggest you find a way to summarise this in a table for both alpha and beta diversity.

Some other comments that I would like to see addressed:

Line 26: confusing “trichoderma” after nitrogen cycling bacteria.

Introduction: maybe add a sentence on what it is WSR? You have this in the methods, but I think you could add this in the introduction. One sentence is enough.

Line 77: maybe reduce the font of the “·” or make it consistent with line 80-81.

Line 85: are these the samples that were already analysed in Yang et al 2019? If they are exactly the same I think you should say this,

Line 101: you can delete the ‘Rhizosphere” at the beginning of the line or make it a subparagraph. Why don’t you do this for the soil as well? Moreover, the methods for bulk soil samples are the same of your previous publication: I think you should mention this.

Line 134 and following: you put the refence of the primers, you can avoid putting the sequences.

If library prep was performed as Yang et al., 2019, please state that.

Table 1 is a bit difficult to read: I do suggest putting this in supplementary and show maybe some boxplots with the only significant differences? If you decide to keep the table, please make it easy to read (introduce lines to separate columns – reduce font). I am also confused by the table content: is it right that at XZ, grain number per plant was 2 in 2017 and 60 in 2018? Was it there an effect of “geography” and “time” as well? I guess the significance levels refers to NSR vs WSR only (but it is just my guess as it is not specified).

You should maybe specify the indoor seed test and yield (all the parameters you tested) in the methods (as they appear in the table 1 then).

Line 206: so you did collapse the years together? Did you prove there was no sign difference in time?

Line 213: comma in the number of fungal OTUs.

You extracted root tissues with a different kit. As you then compare these with other samples (for examples in Fig1, extracted with a different kit) you need to discuss this, as it is well known that different kits affect the microbiome investigation.

Why did you focus only on Shannon? There are other indexes that are commonly used. It would be nice to see if they give similar results.

For beta-diversity: please add the p-values in the text. Moreover, discuss the fact that the samples were extracted with different kits.

I do suggest that you summarise all the beta-diversity data in a table.

Fig 3 is a bit difficult to read. Can you maybe include the treatment in fig 2 and put fig 3 in supplementary?

Paragraph 3.4 why only these two phyla and not all of them?

Line 276: this depletion could be due to the different extraction kit.

Fig 4: the “*” refers to differences between treatment? What about “georgraphy” and “time”?

Line 193: genus level within the 2 phyla you mentioned before of at “whole-community” level?

Fig5: I think colours are already saying if it is down or up

Line 331: there were

Line 370: typo “conclusion”

Line 373: please mention here the effect of kits!

Line 379: I would report here again the R squares.

Line 380: this sentence is difficult to read, please rephrase.

Round 2

Reviewer 1 Report

Thank you for improving the manuscript. The manuscript may be published in the journal Microorganisms.